# Machine learning for Internet of Things (IoT) device identification: a comparative study



Hamid Tahaei[1],*, Anqi Liu[2],*, Hamid Forooghikian[3], Mehdi Gheisari[1], Faiz Zaki[4], Nor Badrul Anuar[4], Zhaoxi Fang[1] and Longjun Huang[5]

[1] Institute of Artificial Intelligence, Shaoxing University, Shaoxing, China
[2] School of Computing and Data Science, Xiamen University Malaysia, Selangor, Malaysia
[3] Department of Management, Islamic Azad University of Kish, Kish, Iran
[4] Department of Computer System and Technology, Faculty of Computer Science and Information Technology, Universiti Malaya, Kuala Lumpur, Selangor, Malaysia
[5] Department of Computer Science and Engineering, Shaoxing University, Shaoxing, China
* These authors contributed equally to this work.

## ABSTRACT

The rapid deployment of millions of connected devices brings significant security challenges to the Internet of Things (IoT). IoT devices are typically resource-constrained and designed for specific tasks, from which new security challenges are introduced. As such, IoT device identification has garnered substantial attention and is regarded as an initial layer of cybersecurity. One of the major steps in distinguishing IoT devices involves leveraging machine learning (ML) techniques on device network flows known as device fingerprinting. Numerous studies have proposed various solutions that incorporate ML and feature selection (FS) algorithms with different degrees of accuracy. Yet, the domain needs a comparative analysis of the accuracy of different classifiers and FS algorithms to comprehend their true capabilities in various datasets. This article provides a comprehensive performance evaluation of several reputable classifiers being used in the literature. The study evaluates the efficacy of filter-and wrapper-based FS methods across various ML classifiers. Additionally, we implemented a Binary Green Wolf Optimizer (BGWO) and compared its performance with that of traditional ML classifiers to assess the potential of this binary meta-heuristic algorithm. To ensure the robustness of our findings, we evaluated the effectiveness of each classifier and FS method using two widely utilized datasets. Our experiments demonstrated that BGWO effectively reduced the feature set by 85.11% and 73.33% for datasets 1 and 2, respectively, while achieving classification accuracies of 98.51% and 99.8%, respectively. The findings of this study highlight the strong capabilities of BGWO in reducing both the feature dimensionality and accuracy gained through classification. Furthermore, it demonstrates the effectiveness of wrapper methods in the reduction of feature sets.

Corresponding author
Longjun Huang, cshlj@usx.edu.cn

## INTRODUCTION

Due to the widespread use of Internet of Things (IoT) devices and their technologies in different sectors, the risk of large-scale cyber-attacks has increased. It is reported that the number of active connections of deployed IoT devices worldwide is 10 billion in 2021, and it is projected that this number will reach 500 billion by 2030 (*Ahmad, Laplante & DeFranco, 2020*). IoT devices include various hardware that can communicate and interact remotely *via* a network. In addition to common smart devices such as PCs, smartphones, and other gadgets, IoT entails a variety of traditional non-smart technologies and everyday objects. These devices connect to and exchange data online through technological integration. However, despite its irrefutable benefits, the growth of IoT poses security and privacy issues because IoT devices lack adequate security designs (*Hao & Rong, 2023*). For instance, because most IoT components have minimal power consumption and limited computational capabilities, they cannot handle sophisticated security mechanisms. In addition, because the majority of IoT devices are wirelessly networked and run without human supervision, they are highly susceptible to hackers (*Abomhara & Køien, 2015*). Consequently, IoT devices are ideal targets for attackers seeking to obtain unauthorized access and infer sensitive information.

Therefore, one of the most crucial tasks for IoT applications and networks is identifying the type of devices connected to prevent potential vulnerabilities, such as cyberattacks on and from IoT devices (*Chakraborty et al., 2021*; *Divakaran et al., 2020*). Consequently, suspicious devices with abnormal activities can be swiftly filtered out by network administrators. Such a procedure is a mitigation process if network administrators know the identities of the infected devices. For example, if a smart bulb is caught by sending emails or streaming films instead of controlling its brightness and color, it can be identified as a suspicious device, and a further filtering process should be performed for this device (*Jmila et al., 2022*).

To solve this problem, the use of hardware and device behavior to identify deployed devices has been extensively studied (*Sanchez et al., 2021*). Depending on the requirements of the environment, behavioral-based identification of IoT devices can be classified into two distinct levels of granularity (*Sanchez et al., 2021*). There are two main approaches: (1) individual device model identification and (2) device type identification.

In individual device model identification, a combination of radio frequency fingerprinting and low-level component analysis is used to distinguish the devices. In low-level component analysis, devices from the same model are differentiated based on hardware manufacturing variations. As an example, the performance of several indicators, such as CPU, GPU, or RAM, is monitored throughout the course of a job. This technique is known as hardware performance analysis (*Sánchez Sánchez et al., 2023*), and is one of the most commonly used approaches for determining device models. To distinguish among devices with the same hardware and software, it is necessary to examine the differences in chip manufacturing, which requires lower-level behavior monitoring for individual device identification (*Sánchez Sánchez et al., 2024*). However, in such cases, attackers can exploit the vulnerabilities in device fingerprinting by modifying specific contextual parameters or hardware-level attributes, which effectively alters the unique characteristics used for

identification. This manipulation can lead to misclassification, ultimately compromising the reliability of the identification system. In the second approach, device type identification (*e.g.*, camera, light bulb) relies on analyzing multiple characteristics, such as network activity patterns and running processes, to classify devices (*Meidan et al., 2017*). By examining the traffic data generated by these devices, machine learning (ML) models can be used to infer their types. If a device exhibits abnormal network behavior that deviates from its expected profile, such anomalies may indicate potential security threats (*Tahaei et al., 2020*). This approach enhances IoT security by allowing administrators to detect and respond to suspicious activities based on behavioral deviations.

Following the trends in the field in recent years, the application of ML techniques, including device identification and fingerprinting, has gained prominence in IoT cybersecurity (*Liu et al., 2022*). Numerous studies have explored ML-based IoT device fingerprinting approaches to achieve a high classification accuracy across different ML models. For example, decision tree (DT) has been widely used for IoT device identification, achieving up to 98% accuracy (*Ammar, Noirie & Tixeuil, 2019*). Similarly, random forest (RF), an ensemble method based on multiple decision trees, has demonstrated a nearly perfect classification accuracy of 99% in IoT device type identification (*Meidan et al., 2017*). Other ML models such as support vector machines (SVM) (*Wanode, Anand & Mitra, 2022*) and neural networks (NN) (*McGinthy, Wong & Michaels, 2019*) have also been applied successfully to IoT security. These solutions typically fingerprint IoT devices by extracting a range of features from the IoT network traffic. For instance, in the case of supervised approaches (*e.g.*, *Sivanathan et al., 2019* and *Thangavelu et al., 2019*), a selective set of features extracted from the network is used to train the classification mode. This process is known as feature selection (FS).

FS algorithms effectively enhance the performance of ML classifiers through data preprocessing techniques (*Shafiq et al., 2021*; *Egea et al., 2018*). These techniques primarily perform statistical calculations to select relevant features and eliminate redundant or duplicate ones, further providing significant benefits (*Wang et al., 2024*). For example, they keep the model from overfitting the training data and raise the accuracy of the classification. Furthermore, FS can preserve the balance between feature count and classification accuracy while lowering the computational complexity and storage requirement. Several FS techniques, such as genetic algorithm (GA) (*Aksoy & Gunes, 2019*), ReliefF (*Zhang, Gong & Qian, 2020*), and NSGA-III (*Du, Wang & Li, 2022*), have been shown to significantly enhance ML performance by optimizing feature set and improving classification efficiency. Specifically, GA-based feature selection in *Aksoy & Gunes (2019)* enabled the classifier to achieve a higher classification rate while using less than half of the original packet header features, demonstrating its effectiveness in improving performance and reducing complexity. The ReliefF algorithm contributed to a classification accuracy in *Zhang, Gong & Qian (2020)* by selecting the most informative features from IoT traffic fingerprints. Also, the NSGA-III-based FS in *Du, Wang & Li (2022)* improved classification accuracy, demonstrating its effectiveness in reducing feature dimensionality while maintaining high accuracy.

In general, there are two types of FS algorithms: filter methods and wrapper techniques. The primary distinction between the two strategies is that wrapper techniques use a classifier to guide FS and assess the feature subset throughout the selection process (*Wang et al., 2024*), while filter methods choose features based on dependence, distance, and information theory (*Hu et al., 2022*). Although wrapper methods provide superior accuracy, they come with increased computational costs due to the iterative nature of feature subset evaluation. Each evaluation requires training and validating a classifier, making wrapper methods significantly more resource-intensive compared to filter methods. However, in applications such as IoT device identification, where accuracy is a critical factor, this trade-off can be justified by the improved classification performance.

While existing solutions have made a profound contribution and proved to be practical and highly accurate in the identification and recognition of IoT devices, a comparative study on different ML classifiers and FS methods is seen as a shortage. Furthermore, there are several surveys, such as *Liu et al. (2022)*, *Mazhar et al. (2021)*, *Sanchez et al. (2021)* that have made an in-depth review of literature; however, a performance analysis of different algorithms in a real-world dataset and a comparative study are neglected. Moreover, some studies, such as *Miettinen et al. (2017)*, *Bezawada et al. (2018)*, *Hamad et al. (2019)* rely solely on expert-extracted features based on their expertise, which may inaccurately represent the flow characteristics exhibited by IoT devices. In addition, due to changes in network settings, non-IP devices might not be accurately identified (*Hamad et al., 2019*), as the value of flow statistics can vary significantly across different network conditions. Besides, some research employs general wrapper FS approaches directly, without taking into account the features of IoT device identification datasets, as in *Kostas, Just & Lones (2022)*, *Aksoy & Gunes (2019)*, and instead just modifies the main heuristic algorithms into binary versions suitable for FS. Moreover, some research on FS for IoT device identification, such as *Chakraborty et al. (2021)*, *Wanode, Anand & Mitra (2022)*, investigated one of the two factors, such as feature subset size or classifier performance, without taking into account the trade-off between the two. Also, the majority of previous studies only use one dataset and lack experimental verification. For instance, *Miettinen et al. (2017)* uses packed-based traffic statistics at the time of IoT device installation, whereas *Bezawada et al. (2018)* simply makes use of a portion of the (*Miettinen et al., 2017*) information. Others such as *Chakraborty et al. (2021)*, *Aksoy & Gunes (2019)*, *Wanode, Anand & Mitra (2022)*, employed only flow statistics-based characteristics, despite the fact that features like flow length and packet count based on flow statistics of IoT devices in various network settings may differ greatly. More recently, the current ML trend is quickly moving towards deep multi-task, multi-modal, federated, deep learning-based, and broad learning solutions. However, some of these solutions are not suitable for the device identification problem due to their low accuracy score, high complexity, large data, and high computational requirements, or they have not been employed on the device identification problem in a supervised learning mode yet.

To fully comprehend the specified difficulties and challenges in IoT device identification with supervised learning approaches, it is necessary to comparatively study FS methods and classifiers with low complexity and comparative performance.

In this study, we argue that the extracted number of selected features is as important as the accuracy of a classifier, as it explicitly affects the performance of a classifier as well as the algorithms' trade-off.

To address this challenge and alleviate its impact, we have adopted the Binary Grey Wolf Optimizer (BGWO), a powerful wrapper-based FS algorithm inspired by the cooperative hunting strategy of grey wolves in nature. This approach enables BGWO to efficiently explore and exploit the search space, dynamically adjusting the FS to optimize classification performance. Unlike conventional wrapper-based methods, BGWO employs an adaptive control mechanism that fine-tunes the balance between exploration and exploitation. This adaptability enhances classification accuracy while effectively reducing the number of selected features, addressing the challenges posed by high-dimensional datasets. These characteristics make BGWO particularly well-suited for IoT device identification, where optimizing computational efficiency without compromising accuracy is essential.

The main contributions of this article are:

- Comparative experimental study: Several highly reputable classical ML classifiers are employed to evaluate their effectiveness and performance.
- Various FS methods: Different methods, including filter-based and wrapper-based, are employed to compare their effectiveness in the accuracy of IoT device detection and analyze trade-offs to find the most suitable FS in each classifier.
- Comprehensive dataset: Two real-world IoT traffic datasets of different types are employed to evaluate the effectiveness and generalization of ML classifiers. Extensive experiments are conducted on these datasets to evaluate the performance of different FS methods.
- Adopting the Binary Grey Wolf Optimizer (BGWO): We have implemented BGWO, a metaheuristic optimization algorithm, as the FS method to find the most optimal feature set that produces the best accuracy. To the best of our knowledge, this is the first attempt to adopt and employ BGWO in the real-world dataset to identify IoT devices.

The remainder of this article is structured as follows. "Related Works" reviews the literature on IoT device identification using ML and FS methods. "Materials and Methods" describes the architecture used in our study, including the data collection and preparation, ML classifiers, FS algorithms, and evaluation metrics. In "Finding and Discussion", the experimental analysis and findings are presented. The remainder of this article is concluded as section.

## RELATED WORKS

IoT device identification is an essential task in the field of network security, as it allows network administrators to detect and monitor the devices connected to their networks. IoT devices need to use network traffic to communicate, and hence, two reputed approaches to fingerprint them include passive and active methods. Active methods involve directly probing the device or network, while passive methods involve observing network traffic

without intruding on the network. In recent years, passive identification has gained popularity due to its ability to automatically monitor traffic data without the need for additional software or hardware and with less intrusiveness compared to the active way.

Traditional methods of passively identifying devices on a network, such as port scanning (*Siby, Maiti & Tippenhauer, 2017*), can be time-consuming and require extensive human resources. ML can be a useful tool for analyzing network traffic because it can be trained on large datasets of network traffic to learn the devices' behaviors and subsequently automatically classify the devices based on the transmitted data from network traffic. Network traffic analysis is widely used in IoT device identification. Researchers study the network traffic generated by IoT devices to understand their behavior and cybersecurity. The communication flow of a device reveals its behavior patterns, making network traffic identification technology a focal point of interest among academia and industry.

*Kawai et al. (2017)* use the support vector machine (SVM) algorithm to identify communication devices by monitoring patterns of traffic. They used only two types of traffic features, *i.e.*, packet size and packet inter-arrival time, to identify six categories of nine devices. By using the SVM, an accuracy of 88.3% in identifying nine devices was obtained. To further improve the identification accuracy, they reused the previously calculated traffic features and added them to the new current traffic features and successfully enhanced the accuracy from 88% to 94%. *McGinthy, Wong & Michaels (2019)* proposed a neural networks (NN)-based SEI algorithm as an additional layer of IoT device and network security to identify IoT devices. The NN-based SEI uses raw in-phase and quadrature (IQ) streams to secure IoT networks. *Ammar, Noirie & Tixeuil (2019)* use a decision tree algorithm to classify 33 IoT devices, including 27 devices from a dataset in *Meidan et al. (2017)*, and six devices from their laboratory. In this work, in addition to the extracted features from the flow characteristics, the textual features from the devices' descriptions were extracted. The textual features are shared in the network payload and presented by a binary bag-of-words model. By combining the two sets of features, they gained an average accuracy of 98%. *Meidan et al. (2017)* used a Random Forest algorithm to analyze labelled network traffic data from seventeen IoT devices representing nine device types to identify authorized devices. On a test set, the trained classifiers detected unauthorized IoT device types with an average accuracy of 96%, while they detected white-listed IoT device types with an almost flawless average accuracy of 99%. In the related study, *Meidan et al. (2017)* conducted an experiment that used GBM, random forest, and XGBoost to construct a multi-stage meta-classifier to identify nine distinct IoT devices. For each of the nine distinct IoT devices, the classifier is used to determine if the device is an IoT or non-IoT device in the first step. The authors then identify certain IoT devices in the second step using the trained classifiers and the relevant parameters they acquired in the first stage, and they achieve a total accuracy of 99.281%. *Miettinen et al. (2017)* proposed IoT SENTINEL, a system by which different types of devices connected to IoT networks are recognized. The system is based on monitoring the communication behavior of devices during the setup process to generate device-specific fingerprints. The authors collected data from 27 IoT devices and extracted 23 features from the flow of

packets generated when the device was connected to the network for the first time. Using the random forest (RF) classifier, findings demonstrated that 81.5% of the devices under consideration could be accurately identified. Similarly, *Sun et al. (2020)* proposed a framework for identifying IoT devices, traffic types, and attacks. They tested various ML algorithms, with the RF classifier achieving the highest accuracy of 99.93% using the extracted features.

ML models are built using a set of features that describe the states of an object (*Chaabouni et al., 2019*). If the features are not enough, the ML model may not have enough information to accurately capture the relationships between the input data and output labels, which can result in underfitting. On the contrary, if the features are too much, the ML model may experience overfitting, which will result in low accuracy and be time-consuming. Many studies perform the FS before constructing their classifiers to improve accuracy and reduce cost. In *Yousefnezhad, Malhi & Främling (2021)*, authors proposed a completely automated single-packet IoT device classifier using different ML algorithms and the genetic algorithm (GA). First, they use GA to select features and then use Decision Table, J48, Decision Trees, OneR, and PART to classify host device types by analyzing features selected by GA. In an experimental study of 23 IoT devices, SysID was able to identify the device type from a single packet with more than 95% accuracy. *Tien et al. (2020)* tried both the unsupervised ML algorithm and supervised learning algorithms to do device identification and anomaly detection. Findings from this study show that the unsupervised learning algorithm, Kmeans, cannot effectively identify IoT devices. By contrast, using supervised learning algorithms has good performance. Artificial neural networks (ANN) obtained 92.8% accuracy in identifying three devices with 110-dimensional features. The result has been further improved to 97.6% after using the gradient-boosted decision trees (XGBoost). The combination of ANN and XGBoost has also achieved a high accuracy of 99.995% in anomaly detection. *Zhang, Gong & Qian (2020)* used the ReliefF FS algorithm to select the important features from both periodic traffic features and protocol features for the K nearest neighbors (KNN) algorithm. They conducted KNN to identify eight devices and gained a correctness of 95% on average. *Sun et al. (2020)* compared the performance of four supervised learning algorithms (K-nearest neighbor, random forest, support vector machine, and multilayer perceptron) on identifying 14 IoT devices collected by *Moustafa (2021)* and eight IoT devices collected by themselves. The study selected 10 features that were calculated from the packet length to make the identification. Results show that Random Forest, which gained an accuracy of 99.5% for the 14 devices and 99.4% for the eight devices, has the best performance. In *Du, Wang & Li (2022)*, used KNN, random forest, and extremely randomized trees (ET) to identify smart home devices. Unlike *Du, Wang & Li (2022)*, which used a feature set combining flow volume, flow rate, and text-based attributes such as port, domain name, and cipher suite, the authors applied a filter-based FS method using NSGA-III to select features exclusively from flow-related statistics. The results demonstrated that the RF algorithm achieved the highest accuracy of 99.5%. Additionally, the NSGA-III-selected feature set proved to be more efficient, reducing both training time and feature dimensionality compared to the feature set used in *Sivanathan et al. (2019)*.

A thorough review of the literature reveals that most studies utilize supervised ML to address various objectives with differing requirements. Consequently, this study focuses on supervised ML in conjunction with several FS algorithms.

## MATERIALS AND METHODS

This article aims to evaluate multiple ML classifiers and the proposed FS method for IoT device identification. The overall architecture involves four steps, including data collection and preparation, ML classifiers, FS methods, and the evaluation phase.

The computing infrastructure for all algorithms and experiments consisted of a device with a 2.6 GHz Intel Core i7 processor, 16 GB of RAM, and the macOS 15.1 operating system. The experiments utilized the R programming language to implement and evaluate the algorithms.

### Data collection and preparation

There are two publicly available datasets employed in this article as follows:

(1) Dataset 1: ProfilIoT dataset (*Meidan et al., 2017*) was designed for IoT device identification and contains network traffic data from eight IoT device types, all connected *via* Wi-Fi, with features extracted from TCP, IP, DNS, and HTTP protocols. The data was collected over several months using a controlled Wi-Fi access point, ensuring consistency. Each captured session was processed into feature vectors representing transport and application layer behaviors. The raw dataset comprises 298 features in CSV format, accessible at https://github.com/Mosseridan/IoT-device-type-identification/tree/master/data. After preprocessing, including normalization and noise removal (irrelevant features or zero-variance features elimination), 188 features remained, which contained 1,600 samples.

(2) Dataset 2: IoT Sentinel (Aalto) the well-known Aalto dataset (*Tien et al., 2020*), a publicly available dataset from Alto University, was developed to enable automated device-type identification for security enforcement in IoT environments. It contains traffic from 31 IoT devices, covering 27 different types, with a subset of 10 devices selected for this study. The dataset was collected by capturing traffic during device setup and regular operation using Wi-Fi, ZigBee, and Ethernet connections. Each device underwent at least 20 resets and reconfigurations to capture diverse traffic patterns. The dataset is available at https://research.aalto.fi/en/datasets/iot-devices-captures. We converted the pcap files collected in *Tien et al. (2020)* to json files to get the names of the features contained in the pcap files. Features are then extracted as the same protocols as in dataset 1 to CSV files. However, we found out that due to the different protocols of devices, features are different in various devices. After eliminating redundant features and identical devices from the same manufacturer that could potentially influence each other, only 70 features from IP and TCP remain, which can be maintained with new devices and functionalities. This process leaves ten unique devices. After normalization for the 70 features and removing the features that are irrelevant or lead to zero variance, 30 features remained, which contain
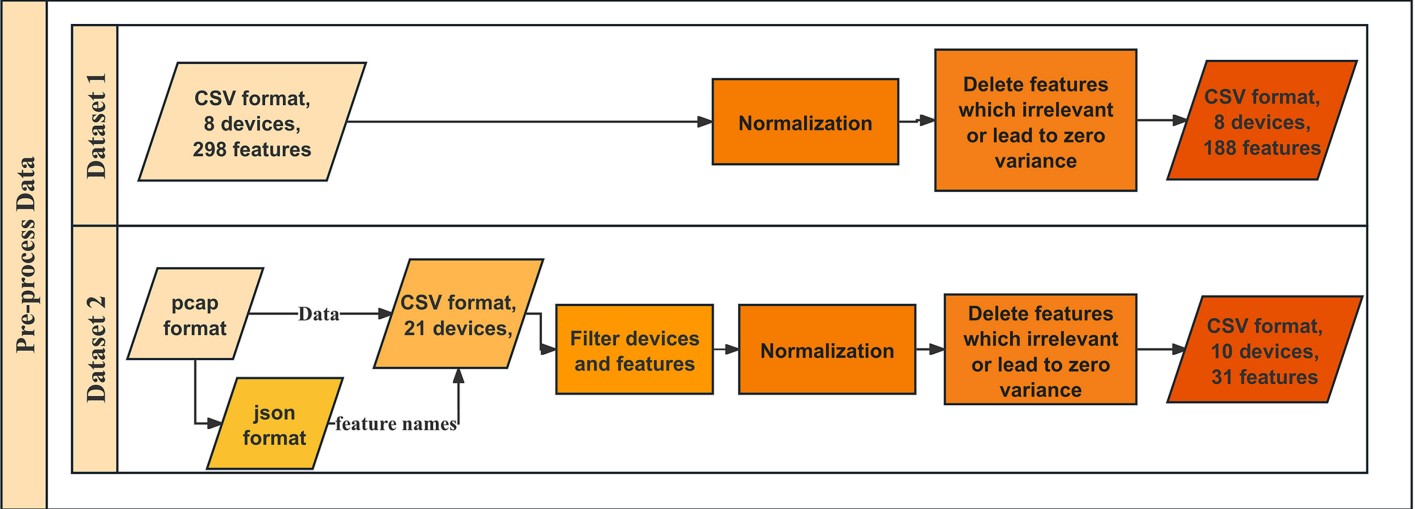

**Figure 1** Workflow of data preprocessing.

**Table 1** Details on the datasets used.

| Item | Dataset 1 | Dataset 2 |
| --- | --- | --- |
| Dataset name | ProfilIoT | IoT Sentinel (Aalto) |
| Source protocol | IP, TCP, DNS, HTTP | IP, TCP |
| Connectivity technology | Wi-Fi | Wi-Fi, ZigBee, Ethernet |
| No. of devices | 8 | 10 |
| No. of samples | 1,600 | 6,000 |
| No. of original features | 298 | 70 |
| No of features after data processing | 188 | 30 |

6,000 samples. Figure 1 shows the workflow of data preprocessing. The detail of the two datasets is shown in Table 1.

## Machine learning classifier

This study evaluated the four most commonly used classical supervised ML classifiers in the literature: (1) Decision tree, a computationally efficient and interpretable classifier, making it a preferred choice in IoT applications (*Ammar, Noirie & Tixeuil, 2019*), (2) Random forest, widely recognized for its high accuracy and robustness in classification tasks. Studies such as *Meidan et al. (2017)* and *Miettinen et al. (2017)* successfully applied RF for IoT device identification, achieving high classification accuracy; (3) SVM, which is frequently used for network traffic classification because of its ability to handle high-dimensional datasets efficiently. *Kawai et al. (2017)* demonstrated that SVM can achieve over 94% accuracy in device identification, and (4) NN, particularly effective for complex pattern recognition. *McGinthy, Wong & Michaels (2019)* showed that NN-based

classification methods could be employed for secure IoT network authentication. Each of these classifiers has demonstrated its effectiveness in IoT device identification and network security applications, with their selection guided by their unique strengths in handling various types of data and classification challenges.

### Decision tree

DT is a supervised learning technique that can be used for both classification and regression tasks. It creates a tree structure that captures each decision taken in the classification or regression as a split in the dataset through nodes and branches. Each internal node is a decision rule, each branch is an outcome, and each leaf node is the final class label or prediction (*Shafiq et al., 2021*). At each step of the process, the best feature is selected based on Gini impurity, entropy, or information gain out of the many features available, which ensures that each the resulting subsets are as pure as possible. One of the main advantages of DT is their interpretability, as they mimic human decision-making processes by providing a clear visual representation of how predictions are made. Additionally, they are computationally efficient and do not require extensive data processing. However, DTs have the major problem of overfitting, which is when an excessively deep trees capture noise rather than general patterns. This is often solved by pruning the trees, combining multiple DTs in the ensemble to increase accuracy and generalization, or setting a maximum depth. Although not without issues, DT is widely used for IoT devices classification, medical diagnosis, fraud detection, and many other areas where simplicity and effectiveness in dealing with struct is required.

### Random forest

RF is a supervised learning algorithm that combines multiple trees through the idea of ensemble learning. An RF model uses the bagging (bootstrap aggregating) method, where each tree is fitted on a random sample of the data, and the final outcome is decided by majority voting (for classification) or averaging (for regression). In addition, RF employs feature randomness by using a random subset of features for every split, which reduces overfitting and enhances generalization. These combination techniques help the RF outperform a single DT by making it less sensitive to problems of high variance, noise, or overfitting. An RF also efficiently manages imbalanced datasets as well as numerical and categorical features. However, its main drawback is higher computational complexity, as training multiple trees requires more processing time and memory compared to a single DT. In IoT device recognition, the models use network traffic features as their communication behaviors for classification purposes. RF can be seen as an ideal candidate due to its high accuracy, noise resistance, and mastery of variable network traffic patterns (*Mahesh, 2018*).

### Support vector machine

SVM, is one of the most often used supervised learning techniques, which is mostly utilized for multi-class and binary classification (*Shafiq et al., 2021*). SVM can be utilized for used for both classification and regression tasks, and it is particularly effective in

high-dimensional spaces. For the classification purpose, it divides the feature space into sub-spaces separated by planes known as hyper-planes. During this division, an "optimal hyperplane" is chosen based on two criteria: it should best separate classes and also maximizes the margin, also known as the distance, between the neighboring data points, or support vectors. Basically, a wider margin ensures a more proficient model. More "non-linearly separable data" poses more challenge, however, it also showcases SVM's strength, through its unique ability to apply different kernel functions, linear, polynomial, and radial basis function (RBF) *etc*. These serve a purpose of transforming the input space into a higher dimensional one and makes it possible for different correspondence classes to become distinguishable. The impressive feature allows SVM to be powerful in computer networking traffic classification, intrusion detection and recognition of IoT devices where complex decision boundaries are essential. Kernel-based methods often show strong classification results, however they come with a downside. The model can be computationally expensive, especially for larger datasets where solving the algorithm becomes a quadratic optimization problem. Nevertheless, SVM remains a widely used and reliable classifier in IoT security applications due to its ability to handle high-dimensional feature spaces and its robustness to overfitting.

### Neural networks

A NN is a mathematical or computer model that simulates the structure and function of a biological neural network. The model is composed of three parts: the input layer, the hidden layer, and the output layer. Each layer consists of interconnected neurons that process and transmit information. The connections between neurons are weighted, and these weights are adjusted during training to optimize the model's performance. The neural network structure used in this study takes various network traffic features as input and processes them through multiple layers to classify different IoT devices. NNs can be employed for both supervised and unsupervised learning tasks. This study utilized a supervised neural network variant. In a supervised NN, the actual output is known in advance. The predicted output is compared to the actual result, and the network parameters are adjusted based on the error. This process is repeated iteratively to improve the accuracy the model (*Mahesh, 2018*).

## Feature selection

This study employs two FS models: (1) Filter-based methods, including the Pearson correlation coefficient (PCC) and mutual information (MI), assess features based on statistical measures, offering a computationally efficient way to select relevant features before model training. These approaches have been widely utilized owing to their scalability and ability to improve the classification accuracy with minimal computational overhead (*Shafiq et al., 2021*; *Chandrashekar & Sahin, 2014*). (2) Wrapper-based methods, such as the Binary Genetic Algorithm (BGA), Binary Particle Swarm Optimization (BPSO), and Binary Grey Wolf Optimizer (BGWO), iteratively evaluate feature subsets based on the classifier performance. These techniques have demonstrated their effectiveness in various studies, consistently improving the classification accuracy

by identifying the most relevant features while minimizing redundant data (*Shafiq et al., 2021*; *Mahesh & Algorithms, 2020*). This section details the FS methods used to optimize the number of selected features.

### Filter-based method

Filtered-based methods are preprocessing tools that evaluate the relevance of each feature independently by ranking features based on certain statistical criteria. The top-ranked features are ten chosen to be used in the ML classifiers. These types of methods are simple, fast, and model-independent.

(a)  Pearson correlation coefficient

Pearson correlation coefficient is a statistical measure that evaluates the linear relationship between two continuous variables. The range of values for PCC is from −1 to +1, where −1 indicates a perfect negative linear correlation, 0 indicates no linear correlation, and +1 indicates a perfect positive linear correlation.

The Pearson correlation coefficient ($\rho$) is calculated by dividing the covariance of the two variables by the product of their standard deviations. The formula to calculate $\rho$ is shown in Eq. (1).

$$\rho_{X,Y} = \frac{cov(X, Y)}{\sigma_X \cdot \sigma_Y} \tag{1}$$

where $cov$ is the covariance, $\sigma_X$ is the standard deviation of $X$ and $\sigma_Y$ is the standard deviation of $Y$.

When two or more features have a high absolute PCC value (*e.g.*, $|r| > 0.9|r| > 0.9|r| > 0.9$), they provide nearly the same information. Keeping both features is unnecessary as one can be removed without significant information loss. PCC aids in dimensionality reduction by identifying and removing highly correlated (redundant) features.

(b)  Mutual information

MI is a measurement of interdependence or association of random variables. It measures the amount of information that can be inferred from one variable by observing the values of another. Mathematically, the MI between two discrete random variables X and Y can be defined as:

$$I(X, Y) = \sum_{x,y} P_{XY}(x, y) \log \frac{P_{XY}(x, y)}{P_X(x)P_Y(y)} \tag{2}$$

where $P_{XY}(x, y)$ is the joint probability distribution function of X and Y, and $P_X(x)$ and $P_Y(y)$ are the marginal probability distribution functions of $X$ and $Y$, respectively. MI quantifies the amount of information one variable provides about the other, with higher mutual information indicating a stronger relationship between the variables. When the two variables are independent, the mutual information is low, reflecting a lack of shared information. Mutual information is computed between each feature and the target variable (device type), capturing the relevance of each feature in identifying the target variable. By

selecting the top-k features with the highest mutual information, the dataset's dimensionality can be reduced while retaining the most informative features for the ML model. While MI measures the mutual dependence between variables which captures both linear and non-linear relationships, PCC measures the linear relationship between variables.

### Wrapper-based method

Unlike the filtered-based types where features are chosen independently, in the wrapper-based approach, features are selected based on data properties and their direct performance on the specific classifiers. Since features are selected dependently, this approach directly optimizes the model and is able to provide superior results compared to filter methods.

However, due to their iterative nature, wrapper-based selection methods require significantly more processing time and computational resources than their counterparts, making them less suitable for real-time or resource-constrained applications. While the improved accuracy is beneficial for IoT device identification, users must consider the trade-off between accuracy and computational efficiency when selecting the features.

(a) Binary Genetic Algorithm

The genetic algorithm is a heuristic optimization algorithm inspired by the process of natural selection and genetics. When used for FS, GA optimizes subsets of features to enhance the performance of a classification model by simulating genetic operations such as selection, crossover, and mutation.

Firstly, an initial population of chromosomes, each representing a feature subset, is generated randomly. Each chromosome can be represented as a binary vector where 1 indicates the feature selected and 0 indicates the feature is not selected. Then, each chromosome's fitness is evaluated using the fitness function which measures the performance of the feature subset in the classification task. Based on fitness values, chromosomes are selected as parents for generating the next generation. The selected parent chromosomes undergo crossover to produce new offspring chromosomes. And offspring chromosomes undergo mutation with a certain probability, where specific genes are randomly flipped. This increases genetic diversity and prevents premature convergence. Then, the new generation of chromosomes replaces the less fit chromosomes from the previous generation. The process is repeated until a stopping criterion is met, such as a predetermined number of iterations or no significant improvement in fitness values.

(b) Particle Swarm Optimization

The PSO algorithm draws inspiration from the social behaviour observed in bird flocking or fish schooling. In this algorithm, each particle corresponds to an individual within the swarm and is characterized by a subset of features, serving as its position in the search space. Initially, a set of particles, representing feature subsets is randomly generated. Subsequently, the velocity and position of each particle are initialized. The fitness of each

particle's position is evaluated using a fitness function, determining the global best position. During each iteration, the velocity and position of each particle are updated based on its personal best position and the global best position until either the maximum number of iterations is reached or there is no significant improvement in the global best fitness value. The velocity update equation is depicted as:

$$v_i(t + 1) = \omega \cdot v_i(t) + c_1 \cdot r_1 \cdot \left(p_i^{best} - x_i(t)\right) \\ + c_2 \cdot r_2 \cdot \left(g^{best} - x_i(t)\right) \tag{3}$$

where, $v_i(t)$ represents the velocity of particle $i$ at time $t$, $x_i(t)$ denotes the position of particle $i$, $p_i^{best}$ is the best position experienced by particle $i$, $g^{best}$ is the best position experienced by all particles, $\omega$ denotes the inertia weight, $c_1$ and $c_2$ are the learning factors, $r_1$ and $r_2$ are random numbers between 0 and 1. In the binary space, the position of the particle consists of 1s and 0s, representing the selection and non-selection of features. Thus, the velocity undergoes a logistic transformation $s(x)$ as shown in Eq. (5), to represent the probability of bit $x_{id}$ taking the value 1. The position of the particle is then updated based on Eq. (4).

$$x_i(t + 1) = \begin{cases} 1, & \text{if } \text{rand}() < s(v_i(t + 1)) \\ 0, & \text{otherwise} \end{cases} \tag{4}$$

$$s(x) = \frac{1}{1 + e^{-x}} \tag{5}$$

(c) Binary Grey Wolf Optimizer

The Grey Wolf Optimizer (GWO) is a metaheuristic optimization algorithm that is inspired by the hunting behaviour of wolves in the wild (*Mirjalili, Mirjalili & Lewis, 2014*).

The wolves are categorized as alpha ($\alpha$), beta ($\beta$), delta ($\delta$), and omega ($\omega$) based on their dominance and leadership. The $\alpha$ is the leader and represents the best solution found, $\beta$ is the best candidate and represents the second-best solution, $\delta$ is on the third level and represents the third best solution, while $\omega$ represents the rest wolves and in the lowest level.

Firstly, the wolves are randomly distributed in the search space and represent candidate solutions, and these positions are updated iteratively to improve the quality of the solutions.

The position of a grey wolf during the hunt is embedded using the mathematical model provided by the following equations.

$$\vec{X}(t + 1) = \vec{X}_p(t) + \vec{A} \times \vec{D} \tag{6}$$
$$\vec{D} = \left| \vec{C} \times \vec{X}_p(t) - \vec{X}(t) \right| \tag{7}$$

where $\vec{X}$ is the position of a grey wolf, $t$ means the number of the current iteration, $\vec{X}_p$ is the position of the prey, $\vec{D}$ is defined in Eq. (2), $\vec{A}$, $\vec{C}$ are generated using the Eqs. (8) and (9).

$$\vec{A} = 2 \times \vec{a} \times \vec{r}_1 - \vec{a} \tag{8}$$

$$\vec{C} = 2 \times \vec{r}_2 \tag{9}$$

where $\vec{r}_1$ and $\vec{r}_2$ are randomly selected from [0,1], $\vec{a}$ is linearly reduced from 2 to 0, which is defined as:

$$\vec{a} = 2 - 2 \times \frac{t}{t_{max}} \tag{10}$$

where the maximum number of iterations is expresses as $t_{max}$.

Usually, the α drives the hunting and β and δ may also engage in hunting. The hunting behaviour of the grey wolves, alpha, beta, and delta, is anticipated to have more information about the location of the prey for the purposes of mathematical modelling.

After the first three candidate solutions are successfully completed, the remaining search agents adjust their positions to match the best search positions of the positions of agents. Equation (11) shows the updated position:

$$\vec{X}(t+1) = \frac{\vec{X}_1 + \vec{X}_2 + \vec{X}_3}{3} \tag{11}$$

where $\vec{X}_1, \vec{X}_2, \vec{X}_3$ are defined in Eqs. (12), (13), and (14) respectively.

$$\vec{X}_1 = \left| \vec{X}_\alpha - \vec{A}_1 - \vec{D}_\alpha \right| \tag{12}$$

$$\vec{X}_2 = \left| \vec{X}_\beta - \vec{A}_2 - \vec{D}_\beta \right| \tag{13}$$

$$\vec{X}_3 = \left| \vec{X}_\delta - \vec{A}_3 - \vec{D}_\delta \right| \tag{14}$$

Here, $\vec{X}_\alpha, \vec{X}_\beta, \vec{X}_\delta$ are the first three best solutions at iteration t, and $\vec{D}_\alpha, \vec{D}_\beta, \vec{D}_\delta$ are defined as follows.

$$\vec{D}_\alpha = \left| \vec{C}_1 \times \vec{X}_\alpha - \vec{X} \right| \tag{15}$$

$$\vec{D}_\beta = \left| \vec{C}_2 \times \vec{X}_\beta - \vec{X} \right| \tag{16}$$

$$\vec{D}_\delta = \left| \vec{C}_3 \times \vec{X}_\delta - \vec{X} \right| \tag{17}$$

In this study, a binary variable from the grey wolf optimizer is utilized for the FS problem. To find a practical solution when BGWO is being executed, the formula for computing the new position, as stated in Eq. (11), is altered, as stated in Eq. (18).

$$Sig\left(\vec{X}(t+1)\right) = \frac{1}{1 + e^{-\vec{X}(t+1)}} \tag{18}$$

The possibility of taking the decision variables is indicated by the value of $Sig\left(\vec{X}(t+1)\right)$, as seen in Eq. (19), where r is a randomly chosen number between 0 and 1.

$$\vec{X}(t+1) = \begin{cases} 1 & if \ r < Sig\left(\vec{X}(t+1)\right) \\ 0 & otherwise \end{cases} \tag{19}$$

In the BGWO a fitness function is required to measure how well a potential solution to a problem performs based on a set of criteria. In the context of FS, it evaluates the effectiveness of a feature set in accurately classifying a target instance. Since we want to find the feature set which has the highest classification accuracy and a minimum number of

selected features. The fitness function used in BGWO to evaluate the feature set used to identify IoT devices using supervised learning algorithms is shown below:

$$fitness\ value\ = \alpha \cdot accuracy + (1 - \alpha) \cdot \frac{|N_o - N|}{|N_o|}\ \alpha \in [0, 1] \tag{20}$$

where *accuracy* is the classification accuracy defined in Eq. (21), the $N_o$ is the length of full feature set before selection, $N$ is the length of the selected feature set, $\alpha$ is the weight of classification accuracy relative to the evaluation criteria and is set to 0.99 in this article.

## Evaluation method

Using a variety of measures, the ML model's performance is assessed during the evaluation phase. This assessment procedure provides insightful information about the efficacy of the model. The following metrics were used in this article:

### *Accuracy of IoT device identification*

Given a dataset comprising IoT device network traffic, the goal is to develop a supervised ML algorithm to accurately identify each device based on its unique device behavior (network traffic). The dataset is separated into a training set and a test set. The training set contained labels used to train the model, and the test set is used to test the classification performance of the model. After the ML model is trained by the labeled training set, it will be used to predict the device label for the label-removed test set. The classification performance of the ML model is measured by accuracy, which is simply a ratio of correctly predicted observations to the total observations and is defined as follows.

$$Accuracy = \frac{TP + TN}{TP + TN + FP + FN}. \tag{21}$$

True/False (T/F) indicates whether the predicted result is right or wrong. Positive/ negative (P/N) represents whether the label of the predicted class is correct or incorrect. Thus, true positive (TP) represents the number of correctly classified positive instances, false positive (FP) represents the number of incorrectly classified negative instances, true negative (TN) represents the number of correctly classified negative instances, and false negative (FN) represents the number of incorrectly classified positive instances.

### *Number of selected features*

FS is a critical task in ML, which aims to select the most informative subset of features that can improve the performance of models. The importance of FS lies in the fact that not all features are equally relevant or useful for the given problem. Using all available features may result in overfitting, while using too few features may lead to underfitting.

The number of selected features can range from the minimum to the maximum of the entire device traffic, which is the number of features extracted from the network traffic. Utilizing all the features may cause computational complexity due to the curse of dimensionality, and irrelevant features may influence the classification accuracy. On the other hand, selecting a few features may result in a loss of information and low accuracy.

Therefore, selecting the optimal subset of features is essential to achieving optimal model performance.

The FS problem can be defined as follow. Select a new subset of features $f' = \{f'_{n_1}, f'_{n_2}, f'_{n_3}, \ldots f'_{n_j}, \ldots f'_{n_{F0}}\}$ from a set of features $f = \{f_1, f_2, f_3, \ldots, f_{F_o}\}$. Where $F$ is the number of selected features and $F_o$ is the number of features in the full feature set before selection. In this article, $F_{o1} = 188$ for the first dataset ($d_1$), and $F_{o2} = 30$ for the second dataset ($d_2$). Since a set containing $F_o$ features can generate $2^{F_o}$ distinct subsets, which is $3.92 \times 10^{56}$ subsets for $d_1$ and $1.07 \times 10^9$ subsets for $d_2$. Therefore, considering a very small dataset containing 1 million records, the total number of distinct subsets for $d_1$, and $d_2$ are $3.92 \times 10^{62}$ and $1.07 \times 10^{15}$ respectively. This is time-consuming to find the best solution, and randomly choosing is not likely to capture the underlying relationships within the data. The main goal of FS is to minimize the number of relevant sets for training and test purposes. Therefore, a heuristic algorithm can give us a feasible solution at an acceptable cost.

In this article, we applied two FS types including filter-based and wrapper-based algorithms. Moreover, a metaheuristic optimizer is used and compared with several other FS methods to identify the best feature subset for minimizing the cost and maximizing the accuracy.

## FINDING AND DISCUSSION

In this article, we allocated 70% of each dataset's samples to create the training set and the remaining 30% to create the test set. To assess the impact of FS, we also performed experiments using the No Feature Selection applied (No-FS) feature set as a baseline and compared the results with those obtained using different FS methods. The experiment was repeated 10 times with different random seeds to ensure reliable outcomes and minimize the effect of random variations. This approach helped us evaluate the performance improvements gained through FS while accounting for any potential biases from the dataset itself.

### Filter-based FS method

#### *Pearson correlation coefficient-based*

We determined the Pearson correlation coefficient (PCC) between each feature and used five distinct cutoff points (0.5, 0.6, 0.7, 0.8, and 0.9) to eliminate features. Every feature whose PCC was higher than the cutoff threshold was eliminated. The dimensionality of the dataset decreased after the features with strong correlation coefficients with other variables were eliminated since those variables might represent the features with high correlation coefficients. The average outcome of employing the PCC in datasets 1 and 2 is displayed in Fig. 2. The number of selected features utilizing the associated boundary is indicated by the purple bar, and the accuracy attained by applying the corresponding classifier and selected features is shown by the lines.
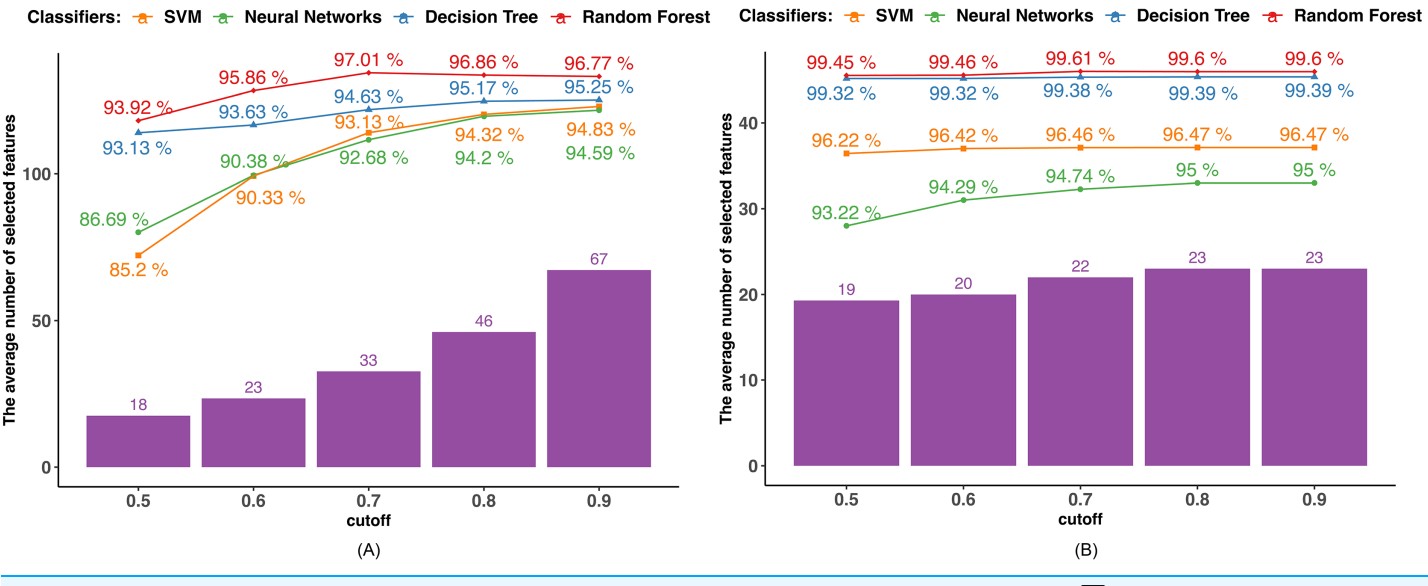

**Figure 2 PCC-based feature selection.** (A) Dataset 1. (B) Dataset 2.

Findings using dataset 1, as shown in Fig. 2A, indicate that only the RF achieved the highest accuracy of 97.01% when the cutoff boundary was set to 0.7, resulting in 33 features being retained. The classification accuracy of the other three classifiers, such as DT, NN, and SVM increased as the cutoff boundary was raised. All of the classifiers reached their highest accuracy with a cutoff boundary of 0.9, retaining 67 features: DT achieved 95.25%, NN achieved 94.83%, and SVM achieved 94.59%. The rate of accuracy improvement gradually slowed as the PCC boundary increased, indicating that features with high PCC have a diminishing or even negative impact on accuracy. For instance, when the number of selected features in the RF classifier increased from 33 to 67, the classification accuracy decreased.

Similarly, in dataset 2, as shown in Fig. 2B, only the RF reaches the highest accuracy of 99.61% when the cutoff boundary is equal to 0.7. Other methods exhibit a better performance in terms of classification as the cutoff border and number of features chosen increase. This is because the majority of the attributes in dataset 2 do not correlate with one another, and the dimensionality is not overly large. As shown in Fig. 2B, 19 out of 30 features have a PCC lower than 0.5. The difference in accuracy caused by the different PCC boundaries is not significant.

### Mutual information-based

Since the mutual information is calculated between two discrete variables, we initially transformed the continuous variables into discrete ones. Each continuous feature value was replaced by the interval to which it belonged after the range of each continuous feature was equally divided into intervals corresponding to the number of devices in the dataset. Next, we determined the mutual information that exists between the target variable and the characteristics. In order to pick features, we examined five MI bounds (0.01, 0.1, 0.2,

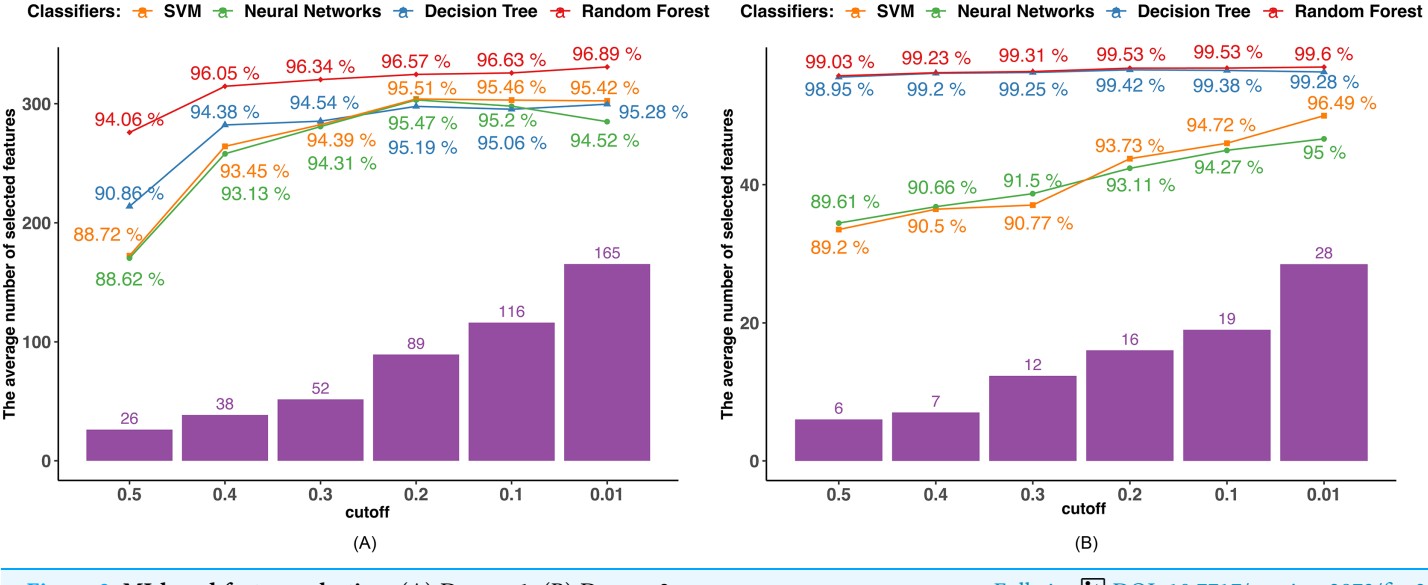

**Figure 3** **MI-based feature selection.** (A) Dataset 1. (B) Dataset 2.

0.3, 0.4, and 0.5) and kept those whose MI values were higher than the border. Figure 3 displays the outcomes of various MI borders.

The performance of all the algorithms improves as the MI boundary increases from 0.5 to 0.6, after which the rate of improvement slows down. The SVM and NN algorithms achieved their highest accuracies of 95.51% and 95.47%, respectively, with an MI boundary of 0.2. In contrast, the tree-based algorithms reached their highest accuracies with an MI boundary of 0.01, achieving 95.42% and 96.89% by DT and RF.

In Dataset 2, DT and RF classifiers demonstrate excellent and consistent performance regardless of the MI boundary selected. Their accuracies reach 98.95% for DT and 99.03% for RF with only six features selected. The accuracies of the SVM and NN classifiers steadily increase as the number of selected features grows. Both classifiers achieve their highest accuracy with an MI boundary of 0.01, with the SVM achieving 95% accuracy and the NN achieving 96.49% accuracy.

## Wrapper-based method

Wrapper-based methods distinguish themselves from filter-based methods by selecting features based on classification outcomes rather than solely on dataset information. Consequently, the features chosen may vary depending on the classifier used. Figures 4 to 7 depict the number of features selected and the classification accuracy achieved by different classifiers across experiments on dataset 1 and dataset 2 using various wrapper-based methods. Each experiment is represented on the horizontal axis, with bars indicating the number of features selected by each FS method for every classifier and lines depicting the corresponding classification accuracies as percentages.

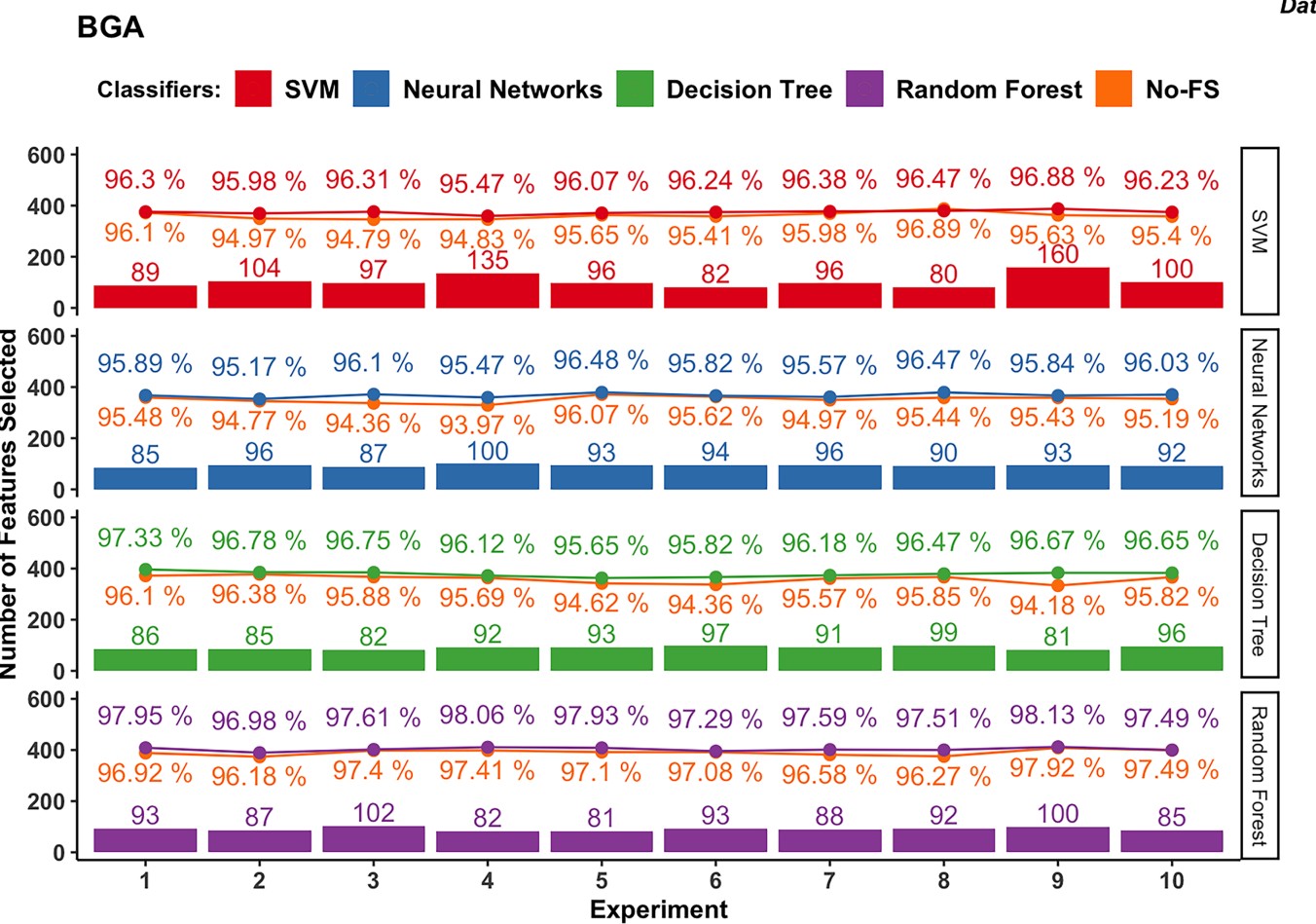

*Dataset1*

Figure 4 **Experiment results of BGA in dataset 1.**

### BGA-based

Figure 4 presents the results from 10 experiments using BGA as a FS method with different supervised learning algorithms.

The number of features selected by the BGA for the SVM classifier varies significantly across experiments, ranging from 80 to 160, as shown in Fig. 4. The classification accuracy achieved by SVM is from 95.47% to 96.88%. The accuracy improvement with BGA with a maximum difference of 1.52% and minimum difference of −0.41% indicates that in some cases, the accuracy was sacrificed for feature reduction. NN exhibits a relatively stable number of selected features, mostly between 85 and 100. The classification accuracy for NN ranges from 95.17% to 96.48%, and the accuracy improvement differs from 0.21% to 1.74%.

Meanwhile, DT classifier in dataset 1 consistently selects between 81 and 99 features. The accuracy remains high, from 95.65% to 97.33%, with an average accuracy of 96.44%. The difference in accuracy improvement is between 0.40% and 2.49%, indicating a significant benefit from FS. Similarly, RF in dataset 1 selects features within a narrow range

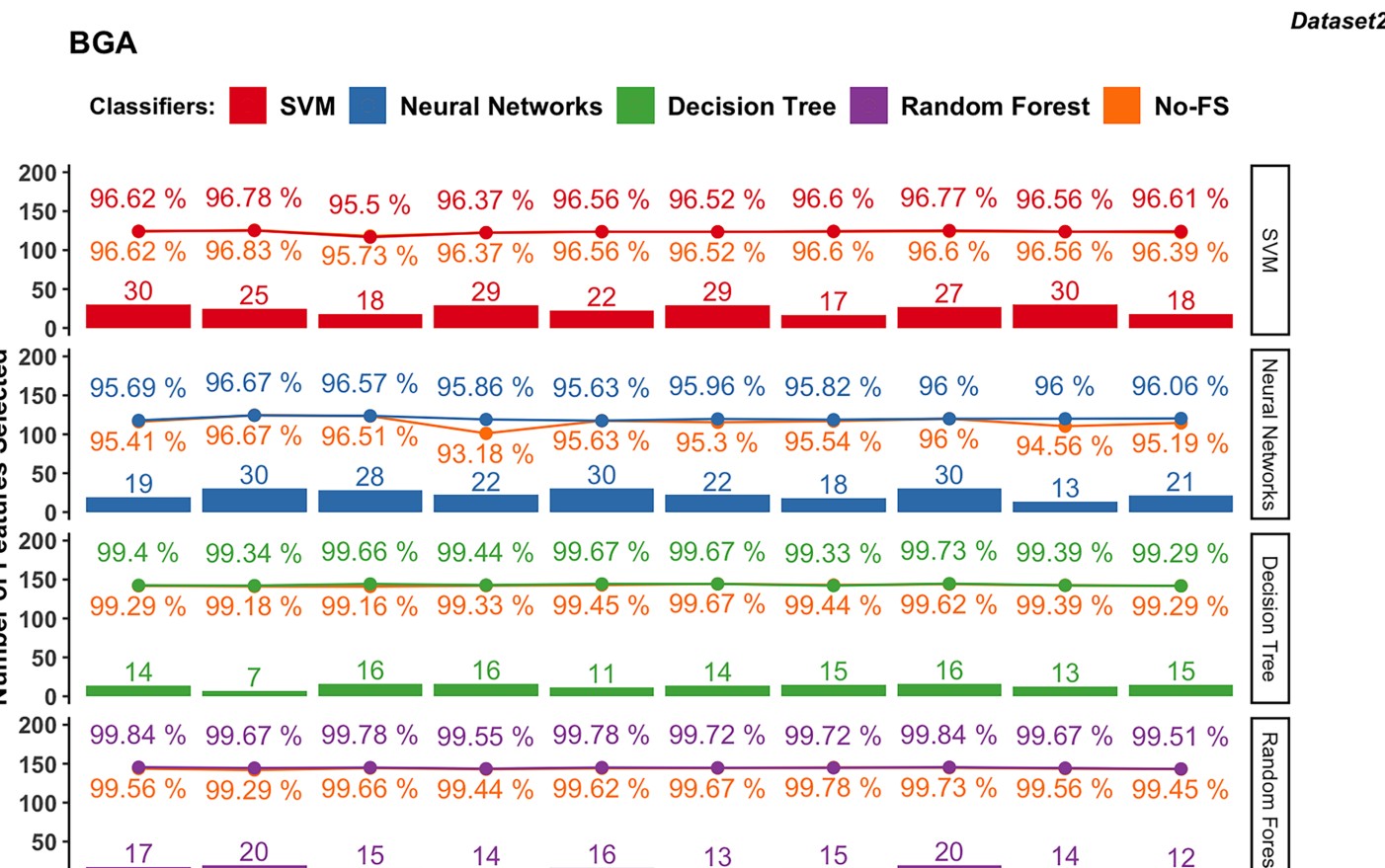

Figure 5 Experiment results of BGA in dataset 2.               

of 81 to 102, averaging at 90.3. The classification accuracy is the highest among the classifiers, ranging from 96.98% to 98.13%, with an average accuracy of 97.65%. The accuracy improvement difference is small, between 0% and 1.24%, demonstrating robustness in performance with selected features.

In dataset 2 shown in in Fig. 5, the SVM classifier still shows a wider range in the number of selected features from 17 to 30, with an average of 24.5. The classification accuracy ranges from 95.5% to 96.78%, with an average of 96.49%. The accuracy improvement is not obvious, ranging from −0.22% to 0.22%. The NN for dataset 2 demonstrates FS variability from 13 to 30, averaging at 23.3. The accuracy ranges from 95.63% to 96.67%, with an average of 96.03%. The improvement difference ranges from 0.00% to 2.68%, indicating effective FS enhancing performance.

The DT classifier in dataset 2 maintains a narrow range of selected features, from 7 to 16, with an average of 13.7. It achieves exceptionally high accuracies between 99.29% and 99.73%, with minimal improvement differences ranging from −0.11% to 0.51%. Similarly, RF in dataset 2 consistently selects between 12 and 20 features, achieving accuracies from

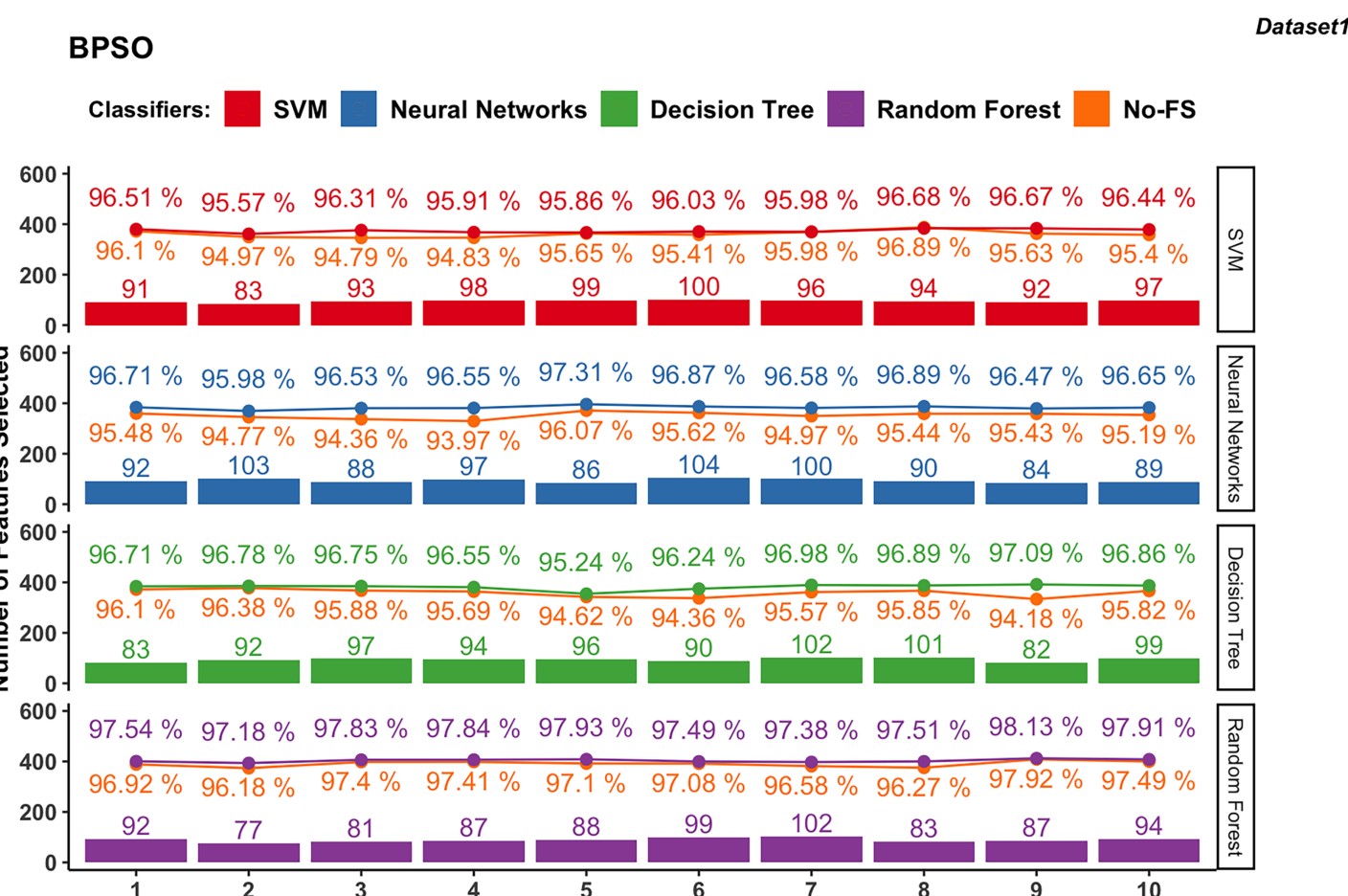

**Figure 6 Experiment results of BPSO in dataset 1.**

99.51% to 99.84%. The improvement difference remains minimal, ranging from −0.06% to 0.38%, indicating stable performance with the selected feature set.

### BPSO-based

In dataset 1, the features selected by BPSO for SVM classifier ranges from 83 to 100, with an average of 94.3 as shown in Fig. 6. The classification accuracy varies from 95.57% to 96.68%, averaging at 96.20%. The improvement in accuracy ranges from −0.21% to 1.52%, indicating that while FS can slightly decrease accuracy, it generally provides a moderate improvement.

In dataset 1, the NN using BPSO ranges between 84 and 104 features, with an average of 93.3. The accuracy ranges from 95.98% to 97.31%, with an average of 96.65%. The accuracy improvement difference ranges from 1.04% to 2.59%, showing a significant benefit from FS and consistently enhancing performance.

The DT classifier in dataset 1, employing BPSO, selects features within a narrow range of 82 to 102, averaging 93.6. The classification accuracy ranges from 95.24% to 97.09%,

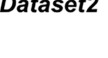

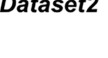

**Figure 7  Experiment results of BPSO in dataset 2.**

with an average of 96.61%. The accuracy improvement difference ranges from 0.40% to 2.91%, indicating substantial gains from FS.

The RF for dataset 1, using BPSO, consistently selects features between 77 and 102, averaging 89. The classification accuracy is high, ranging from 97.18% to 98.13%, with an average of 97.68%. The accuracy improvement difference is smaller, between 0.21% and 1.24%, demonstrating stable performance with the combination of BPSO and RF.

The NN in dataset 2 (Fig. 7), using BPSO, ranges from 14 to 30, averaging 18.9. The accuracy varies from 95.09% to 96.51%, with an average of 96.01%. The accuracy improvement difference ranges from −0.33% to 2.63%, suggesting that FS with BPSO can significantly enhance performance for NN, though there are some cases of slight decreases.

In dataset 2, DT with BPSO shows a narrow range of selected features, from 10 to 30, averaging 15.7. The accuracy is consistently high, ranging from 99.12% to 99.73%, with an average of 99.39%. The accuracy improvement difference is minimal, from −0.16% to 0.22%.

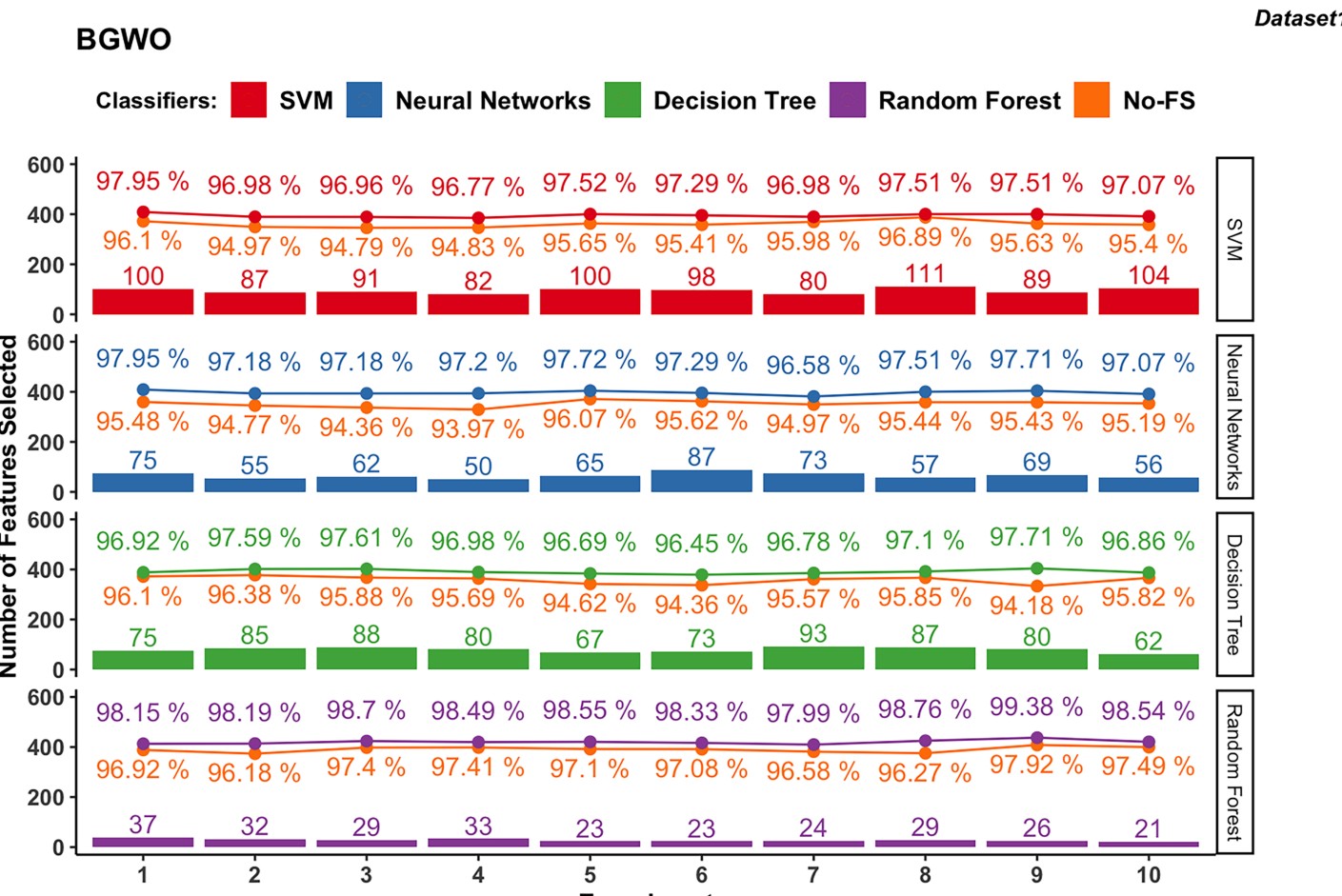

**Figure 8  Experiment results of BGWO in dataset 1.**     

The RF with BPSO selects features between 9 and 17, with an average of 13.6. The classification accuracy ranges from 99.45% to 99.89%, with an average of 99.71%. The accuracy improvement difference is small, from −0.05% to 0.34%. Since DT and RF already have great performance using the No-FS in Dataset 2, the accuracy improvement of FS is not obvious, but the effect of dimensionality reduction is considerable.

### BGWO-based

Figure 8 illustrates the results obtained from employing BGWO across various supervised learning algorithms in dataset 1. The classification accuracy of the SVM algorithm increased from 94.83% to 96.89% using the No-FS feature set to 96.96% to 97.95% by using the BGWO selected feature set. The range of accuracy increment is 0.62–2.17%, which is very considerable. Meanwhile, the 80 to 111 features are selected from the 188 features by BGWO for the SVM classifier.

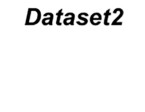

Figure 9 Experiment results of BGWO in dataset 2.

Similarly, NN in dataset 1 saw notable accuracy improvements. Initially ranging from 93.97% to 96.07%, accuracies improved to 96.58% to 97.95% after using BGWO-selected features, with improvements ranging from 1.61% to 3.23%. Compared with SVM, the number of selected features using NN is less, from 50 to 87.

The range of accuracy using the DT is 96.18–97.92% using the No-FS feature set and 96.45–97.71% using the BGWO, with improvements ranging from 0.11–0.5%. The number of selected features for the decision tree is from 62 to 93, which is between neural networks and SVM. For another tree-based algorithm, since RF combines multiple decision trees, it has the advantages of being less prone to overfitting and better performance in handling high-dimensional feature spaces. The classification accuracy using RF increased from 96.18–97.92% to 97.99–99.38% after using BGWO, which is the highest among all algorithms. BGWO reduced 188 features to 21–37 features for RF, demonstrating a superior reduction in time complexity.

In contrast, dataset 2 showed less pronounced improvements with SVM, where accuracies ranged from 95.73% to 96.83% with the No-FS feature set and 96.01% to 96.99% with selected features (Fig. 9), with minor improvements from −0.16% to 0.39%. The NN varied widely in accuracy due to data sensitivity, ranging from 93.18% to 96.67% initially and improving to 95.64% to 98.52% with BGWO, showcasing accuracy improvements of 0.66% to 2.52% with 13 to 23 features selected out of 30.

Both DT and RF in dataset 2 maintained steady performance, with DT achieving accuracies from 99.16% to 99.67% initially to 99.38% to 99.84% with selected features. RF achieved accuracies from 99.29% to 99.78% initially to 99.62% to 99.95% with selected features, showing minimal accuracy increments from 0.11% to 0.50% for DT and 0.00% to 0.38% for RF. DT selected 13 to 23 features, while RF selected 5 to 15 features, underscoring the efficient FS capabilities of BGWO for them.

## Comparison between FS methods

In order to achieve the best performance in both FS and classification accuracy, we calculate the fitness value $\left( \textit{fitness value} = 0.99 \cdot \textit{accuracy} + 0.01 \cdot \dfrac{|N_o - N|}{|N_o|} \right)$ which used in wrapper-based methods also for MI-based and PCC-based methods to evaluate the performance of each boundary. The results of the boundary used to achieve the highest fitness value are chosen to represent the FS method for comparison. Figures 10 and 11 illustrate the comparative results among different classifiers using various FS methods.

### *Supervised classifiers*

In dataset 1 shown in Fig. 10, the RF consistently demonstrates the highest accuracy across all FS methods, particularly excelling with BGWO. It performs exceptionally well on both high-dimensional and low-dimensional datasets, showcasing its robust nature. The other three classifiers do not exhibit significant differences in recognition accuracy.

The DT algorithm shows slightly better recognition accuracy than SVM and NN when using PCC, BGA, and BPSO methods. However, NN did not perform well with other FS algorithms but showed strong performance with the BGWO method. This significant improvement indicates that BGWO's effective FS greatly enhances the learning capability of NN. SVM displays moderate accuracy and, when combined with different FS methods, consistently selects a larger number of features, confirming its better performance on high-dimensional datasets.

In dataset 2 (Fig. 11), DT and RF exhibit significantly higher accuracy compared to SVM and NN. The DT performs notably better in dataset 2, which contains 30 features, than in dataset 1, which contains 188 features. DT performs better on lower-dimensional data because such datasets can more effectively prevent overfitting in DT.

The RF continues to demonstrate the highest accuracy regardless of the FS method used. Since both DT and RF achieved exceptional performance on the No-FS feature set, the FS algorithms do not significantly improve their recognition accuracy. However, these algorithms effectively reduce the number of features without compromising accuracy. The lower dimensionality of dataset 2 imposes some limitations on the effectiveness of FS

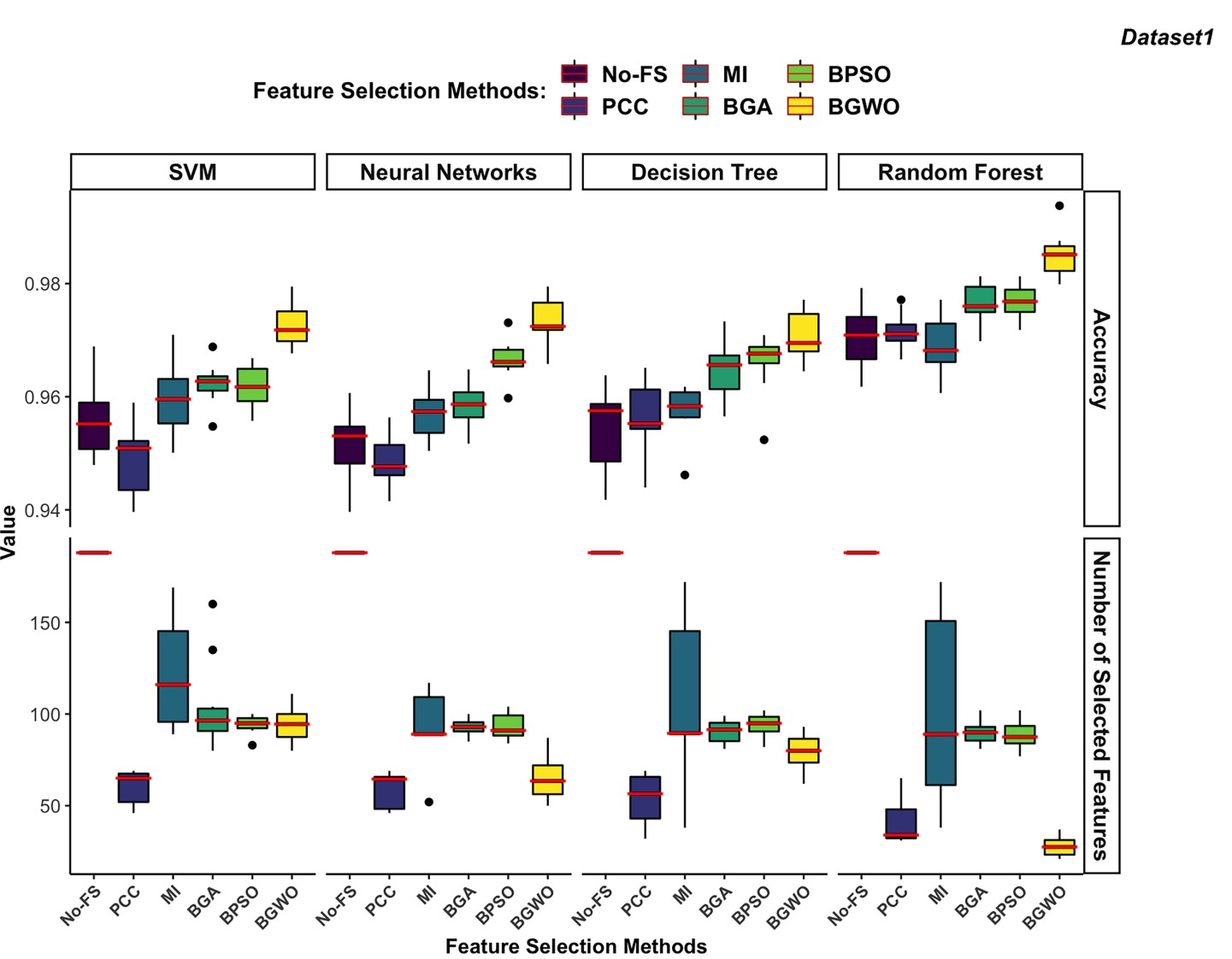

**Figure 10 Comparison between FS methods and classifiers in dataset 1.**

algorithms. Additionally, SVM's preference for high-dimensional datasets means its recognition accuracy is not outstanding in dataset 2, and FS does not significantly improve its performance. Consistent with its performance in dataset 1, SVM performs better on high-dimensional data, which is why the FS algorithms result in fewer feature reductions.

The NN shows similar behavior in dataset 2 as in dataset 1, with a notable improvement in recognition accuracy only when using the BGWO algorithm. This indicates the irreplaceable role of the BGWO method in enhancing NN's ability to select features for IoT device recognition.

### Feature selection methods

Filter-based algorithms select features based on statistical relationships between features or between features and the target variable, whereas wrapper-based algorithms choose

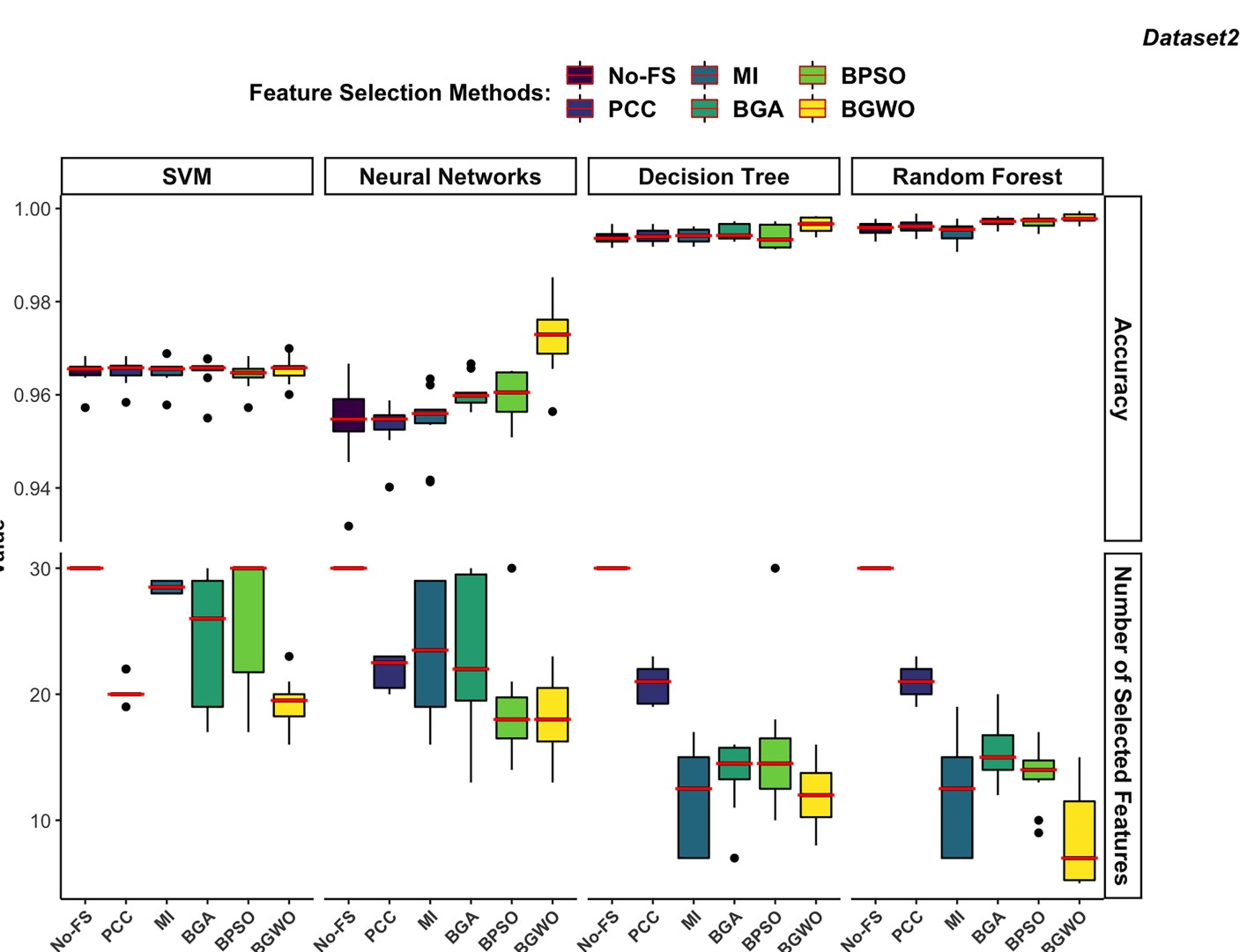

**Figure 11 Comparison between FS methods and classifiers in dataset 2.**

features through iterative training and evaluation of the model. In both dataset 1 and dataset 2, wrapper-based methods consistently demonstrated higher performance in improving recognition accuracy compared to filter-based methods.

Filter-based methods often fail to reduce features without compromising accuracy. As illustrated in Fig. 10, the PCC algorithm significantly reduced the number of features in dataset 1 but also noticeably decreased the recognition accuracy for SVM and NN. This suggests that PCC may have removed features that are highly relevant for these classifiers, leading to a drop in performance. Unlike wrapper-based methods, which evaluate feature importance based on model training, PCC relies on correlation coefficients, which do not always capture the most predictive features for ML models. Similarly, the RF algorithm performed slightly worse on the selected features by the MI-based method compared to the No-FS feature set. This indicates that MI might not always be the optimal choice for FS, as

it focuses on information gain without considering the interdependencies among multiple features simultaneously. Among filter-based methods, the MI-based method outperformed the PCC-based method for SVM, NN, and DT classifiers in dataset 1. However, PCC proved more effective for the RF classifier.

In contrast, wrapper-based methods managed to reduce the number of features without degrading recognition accuracy. Among these methods, BGWO consistently exhibited superior performance across all classifiers and datasets. Using the same number of search agents and iterations, BGWO selected fewer features while achieving higher accuracy for each classifier. Although BGA and BPSO were slightly less effective than BGWO, they still significantly reduced the number of features and improved recognition accuracy. In dataset 1, BPSO and BGA selected a comparable number of features for each classifier, with BPSO generally providing better accuracy improvements. BPSO enhanced the precision of NN, DT, and RF classifiers more effectively than BGA, which only slightly outperformed BPSO for the SVM algorithm. In a similar vein, these findings align with previous studies that have demonstrated the effectiveness of wrapper-based FS techniques in enhancing classification performance. Studies such as *Aksoy & Gunes (2019)*, *Zhang, Gong & Qian (2020)*, and *Du, Wang & Li (2022)*, reviewed in the literature section, report similar trends where FS methods significantly optimize ML performance while reducing feature dimensionality. For instance, the GA-based FS approach in *Mazhar et al. (2021)* enabled classifiers to achieve high accuracy while reducing feature sets by more than half, minimizing computational complexity. Similarly, ReliefF in *Zhang, Gong & Qian (2020)* improved classification accuracy to 95% by selecting the most informative IoT traffic features, and NSGA-III in *Du, Wang & Li (2022)* enhanced classification accuracy from 94.5% to 95.8%.

In dataset 2, which has fewer features, FS algorithms did not significantly enhance the recognition accuracy of SVM, DT, and RF algorithms. However, it was still observed that wrapper-based methods outperformed filter-based methods, as shown in Fig. 11. BGWO remained the best-performing FS algorithm, not only selecting a more concise feature set to improve recognition efficiency but also enhancing recognition accuracy. The inferior performance of PCC and MI in dataset 2 further confirms that these filter-based methods may not be suitable for scenarios where complex feature interactions play a critical role. Since they do not consider feature dependencies holistically, they may discard relevant features, leading to suboptimal classifier performance. The NN demonstrates more sensitivity to FS compared to other classifiers and shows significant performance improvement with BGWO, similar to its performance in dataset 1. BGWO improved NN by selecting the most relevant features, significantly enhancing its performance. Similarly, BPSO and BGA improved NN's performance to some extent, whereas PCC and MI did not contribute to any increase in classification accuracy for NN.

Findings from our experiments align with our study, confirming that FS techniques consistently enhance classification performance across various datasets and methodologies. The consistency of our results with prior studies validates the effectiveness of our approach and demonstrates its generalizability to different domains.

# CONCLUSION

This article presented a performance evaluation of several high-reputation ML classifiers for IoT device identification. We assessed the effectiveness of network traffic features by leveraging various FS methods comprising filter-based and wrapper-based approaches in fitting the best model and achieving optimal accuracy across different classifiers. Additionally, the study introduced the BGWO to identify the optimal subset of features for IoT device identification. Two publicly available IoT network traffic datasets were utilized to ensure the reliability of our findings. Our analysis indicates that wrapper-based FS methods outperform filter-based methods in terms of accuracy and feature reduction. BGWO proved to be the most effective FS method, which significantly enhanced the performance of RF and NN classifiers. Compared to other wrapper-based methods like BPSO and BGA, the BGWO selected fewer features while achieving higher accuracy across all classifiers.

## Limitations and future work

The limitations of this article include the ongoing open issues pertaining to the privacy of data, including evaluating the current ML trends, such as multimodal and federated-based learning. Also, to gain more reliable outcomes, we believe that larger datasets comprising more devices with more features can reveal deeper insights into the behavior of classifiers and FS methods. This study focused on comparing filter-based and wrapper-based approaches with supervised classifiers. In future work, we plan to explore combining BGWO with unsupervised and deep learning-based algorithms to further enhance IoT device identification.

## Funding

This work was supported by the Natural Science Foundation of Zhejiang Province under grant no. LGF22F010006, and the Humanities and Social Science Research Project of Ministry of Education of China under grant no. 22YJAZH016. The funders had no role in study design, data collection and analysis, decision to publish, or preparation of the manuscript.

## Grant Disclosures

The following grant information was disclosed by the authors:
Natural Science Foundation of Zhejiang Province: LGF22F010006.
Humanities and Social Science Research Project of Ministry of Education of China: 22YJAZH016.

## Competing Interests

The authors declare that they have no competing interests.

## Author Contributions

- Hamid Tahaei conceived and designed the experiments, analyzed the data, prepared figures and/or tables, authored or reviewed drafts of the article, and approved the final draft.
- Anqi Liu conceived and designed the experiments, performed the experiments, analyzed the data, performed the computation work, prepared figures and/or tables, and approved the final draft.
- Hamid Forooghikian analyzed the data, prepared figures and/or tables, authored or reviewed drafts of the article, and approved the final draft.
- Mehdi Gheisari analyzed the data, authored or reviewed drafts of the article, and approved the final draft.
- Faiz Zaki analyzed the data, authored or reviewed drafts of the article, and approved the final draft.
- Nor Badrul Anuar analyzed the data, authored or reviewed drafts of the article, and approved the final draft.
- Zhaoxi Fang analyzed the data, prepared figures and/or tables, authored or reviewed drafts of the article, and approved the final draft.
- Longjun Huang analyzed the data, prepared figures and/or tables, authored or reviewed drafts of the article, and approved the final draft.

## Data Availability

The ProfilIoT Dataset (Dataset 1) is available at GitHub: https://github.com/Mosseridan/IoT-device-type-identification/tree/master/data.

The IoT Sentinel (Aalto) Dataset (Dataset 2) is available at: https://research.aalto.fi/en/datasets/iot-devices-captures.

## Supplemental Information

Supplemental information for this article can be found online at http://dx.doi.org/10.7717/peerj-cs.2873#supplemental-information.

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
