# Peer review of "Machine learning for Internet of Things (IoT) device identification: a comparative study"

_PeerJ Computer Science, doi:10.7717/peerj-cs.2873_

## Round 0.1 · original submission · Major Revisions

I have received reviews of your manuscript. The reviewers have found the topic interesting; however, several concerns regarding experimental setup, results, and presentation must be addressed. These issues require a major revision. Please refer to the reviewers’ comments at the end of this letter; you will see that they advise you to revise your manuscript. If you are prepared to undertake the work required, I would be pleased to reconsider my decision. Please submit a list of changes or a rebuttal against each concern when you submit your revised manuscript.

Thank you for considering PeerJ Computer Science for the publication of your research.

With kind regards,

Reviewer 1 ·

Basic reporting

The title of the manuscript is " Machine Learning for IoT Device Identification: A Comparative Study". The introduction clearly identifies the motivation and relevance of the study, particularly emphasizing the challenges in IoT security and the importance of device fingerprinting using ML. The paper addresses a critical gap by conducting a comparative analysis of classifiers and feature selection methods, which adds value to the domain. Including binary green wolf optimizer (BGWO) as a novel component highlights the innovation in the study. The use of two widely utilized datasets to validate findings ensures reliability and generalizability.

Experimental design

The paper addresses a critical gap by conducting a comparative analysis of classifiers and feature selection methods, which adds value to the domain. Including binary green wolf optimizer (BGWO) as a novel component highlights the innovation in the study.

Validity of the findings

The use of two widely utilized datasets to validate findings ensures reliability and generalizability.

Additional comments

The title of the manuscript is " Machine Learning for IoT Device Identification: A Comparative Study". The introduction clearly identifies the motivation and relevance of the study, particularly emphasizing the challenges in IoT security and the importance of device fingerprinting using ML. The paper addresses a critical gap by conducting a comparative analysis of classifiers and feature selection methods, which adds value to the domain. Including binary green wolf optimizer (BGWO) as a novel component highlights the innovation in the study.The use of two widely utilized datasets to validate findings ensures reliability and generalizability. The suggestions for improving the manuscript is listed as follows:
"Identification of devices" could be more precisely rephrased as "IoT device identification."
Ensure consistent capitalization of terms like "Feature Selection" (FS) and "Machine Learning" (ML) throughout the paper.
Provide brief descriptions of the datasets used (e.g., size, characteristics, or application domains) to help readers understand the context of the evaluation.
While BGWO is introduced, its working mechanism and novelty compared to other optimizers could be briefly explained in the introduction for better clarity.
Add specific numerical results or comparative performance metrics in the abstract/intro to give readers an early insight into the study’s key findings.
While the text mentions the effectiveness of wrapper methods in reducing feature sets, the contribution could be elaborated. For example, highlight the computational cost or trade-offs involved.
Mention why the selected classifiers and feature selection methods are "reputable" or significant in the literature to provide context for the study.
Reorganize sentences for smoother readability. For example: Replace: "It further examines the efficacy of different FS methods including filter and wrapper-based techniques across these ML classifiers." With: "The study evaluates the efficacy of filter- and wrapper-based feature selection methods across various ML classifiers."
The introduction could benefit from briefly citing key related works to support claims about the accuracy of ML and FS methods for IoT security.
"being regarded" should be rephrased for grammatical correctness, e.g., "and is regarded as."
The limitations and future works must be given.

Reviewer 2 ·

Basic reporting

The paper presents a generally clear narrative, but there are areas where the language could be improved for clarity and precision. For instance, the explanation of how an attacker can manipulate the device fingerprint and the implications for IoT device identification (lines 70-72) lacks clarity. It is not clear whether the attacker alters the fingerprint or if the identification method fails to work as intended. Additionally, the feature selection process described in lines 324-326 raises questions about the methodology, e.g. the potential for features to correlate with only subsets of other features. This could lead to an incomplete understanding of the relationships among features. Overall, while the results are presented clearly, the paper would benefit from a more rigorous and precise use of language throughout.

Experimental design

The research is original and falls within the Aims and Scope of the journal. The research question is well-defined and addresses a relevant knowledge gap in the field of IoT security. The investigation appears to be rigorous, adhering to high technical and ethical standards.

Validity of the findings

While the results are presented in a clear and comprehensive manner, the impact and novelty of the findings are not thoroughly assessed. The accuracy numbers across different threshold cutoffs and parameter settings show only minor variations, raising concerns about the significance of the improvements observed. The overall accuracy of the models lies between 96%-99% predominantly. The impressive accuracy may be due to the effective feature selection method; however, it is important for the authors to compare the models without feature selection to ensure that the positive results are not merely a reflection of dataset quality. It is crucial for the authors to address whether these improvements are significant and generalizable. The conclusions drawn are generally well-stated and linked to the original research question, but further validation of the findings is necessary to strengthen the paper's contributions to the field.

Additional comments

The paper demonstrates a solid foundation in its exploration of machine learning models for IoT security, with clear and informative result figures. However, the inclusion of basic examples of decision trees, random forests, support vector machines, and neural networks (figures 2-5) may not be appropriate for a research paper and could be replaced with more detailed presentation of the final trained models. This would enhance the paper's contribution to the field.

---

## Round 0.2 · accepted · Accept

All reviewers' concerns have been thoroughly addressed, so I am pleased to inform you that your work has now been accepted for publication in PeerJ Computer Science.

Thank you for submitting your work to this journal. I look forward to your continued contributions on behalf of the Editors of PeerJ Computer Science.

With kind regards,

Reviewer 2 ·

Basic reporting

The author has effectively addressed all previous concerns.

Experimental design

The author has effectively addressed all previous concerns.

Validity of the findings

The author has effectively addressed all previous concerns.

Additional comments

I would like to express my appreciation for the author's efforts in addressing the previous comments. The revisions have significantly improved the manuscript.